# Towards a Flexible Assessment of Compliance with Clinical Protocols Using Fuzzy Aggregation Techniques

Anna Wilbik [1,*], Irene Vanderfeesten [2], Dennis Bergmans [3,4], Serge Heines [3], Oktay Turetken [5] and Walther van Mook [3,6]

1   Department of Advanced Computing Sciences, Maastricht University, 6229 EN Maastricht, The Netherlands
2   Research Centre for Information Systems Engineering, Faculty of Economics and Business, KU Leuven, 3000 Leuven, Belgium
3   Department of Intensive Care Medicine, Maastricht University Medical Centre+, 6229 HX Maastricht, The Netherlands
4   School of Nutrition and Translational Research in Metabolism (NUTRIM), Maastricht University, 6229 HX Maastricht, The Netherlands
5   Department of Industrial Engineering and Innovation Sciences, Eindhoven University of Technology, 5612 AZ Eindhoven, The Netherlands
6   Academy for Postgraduate Training, Maastricht University Medical Centre+, School of Health Professions Education, Maastricht University, 6229 ER Maastricht, The Netherlands
*   Correspondence: a.wilbik@maastrichtuniversity.nl

**Abstract:** In healthcare settings, compliance with clinical protocols and medical guidelines is important to ensure high-quality, safe and effective treatment of patients. How to measure compliance and how to represent compliance information in an interpretable and actionable way is still an open challenge. In this paper, we propose new metrics for compliance assessments. For this purpose, we use two fuzzy aggregation techniques, namely the OWA operator and the Sugeno integral. The proposed measures take into consideration three factors: (i) the degree of compliance with a single activity, (ii) the degree of compliance of a patient, and (iii) the importance of the activities. The proposed measures are applied to two clinical protocols used in practice. We demonstrate that the proposed measures for compliance can further aid clinicians in assessing the aspect of protocol compliance when evaluating the effectiveness of implemented clinical protocols.

**Keywords:** clinical protocols; protocol compliance; protocol adherence; protocol conformance; aggregation; OWA operator; Sugeno integral





## 1. Introduction

Recent developments in medicine have resulted in better diagnostic methods and increased number or improved efficacy of treatments, further resulting in improved quality of care at the cost of more expensive and complex healthcare. In order to deal with this complexity and anticipate the increased demand for care, many hospitals and healthcare providers standardize care processes by using clinical guidelines and protocols [1]. Such an approach has many benefits [2,3], such as reduced hospital complication rates without an increased length of stay [4] and reduced treatment costs [5].

Field and Lohr [6] define a *clinical guideline* as "systematically developed statements to assist practitioners and patient decisions about appropriate health care for specific circumstances". Clinical guidelines are designed based on the best available evidence and can be seen as a set of rules that do not need to be strictly followed since individual patient variation may necessitate deviation from such guidelines. In order to make clinical guidelines more operational, many hospitals develop more specific clinical protocols for specific, more standardized care processes in which limited variation in care is known to exist. *Clinical protocols* provide "a comprehensive set of rigid criteria outlining the management steps for a single clinical condition or aspects of organization" [7]. Hospitals

can have many of such clinical protocols implemented in daily practice [8], e.g., clinical protocols for detoxification [9], or the glucose management or weaning protocols as used in many intensive care units (ICUs) [10].

The effect of applying clinical protocols in daily practice, in general, is measured during intervention case studies [11,12]. In such studies, a situation is assessed before and after the introduction of the protocol. Moreover, compliance with the protocol is measured and reported as evidence that the observed outcome change was indeed caused by following the protocol. In this context, by *compliance*, we mean "the degree to which the behavior of the executors of the clinical protocol corresponds to the behavior described in the clinical protocol" [13]. In the literature (and in this paper), the terms *adherence*, *compliance* and *conformance* are used interchangeably [14–16]. In the medical domain, however, the term adherence is sometimes preferred (e.g., patient adherence) because compliance has a negative connotation [17,18].

Measurement of compliance is not a part of the routine in daily practice, however. Therefore, it is difficult to quantify the effectiveness of a clinical protocol on a daily basis. One of the difficulties with automatically measuring compliance is that not every activity is recorded in the hospital information system. Information about some performed activities, e.g., whether a certain test was performed, if not entered manually or automatically, nevertheless could be deduced from the presence of specific patients' results' in the system. Another difficulty is that a widely accepted definition of compliance or compliance metrics is lacking. Two commonly used compliance metrics are (1) the percentage of protocol executions with all activities compliant and (2) the percentage of compliant activities (over all patients) for a given protocol [11,12]. In those approaches, it is assumed that a single activity within the protocol can be either compliant with the clinical protocol or not compliant. However, when discussing with healthcare providers, especially more experienced doctors and nurses, one can notice that compliance is not a binary state. For instance, some thresholds are too strict for specific patients who require a different treatment strategy. Therefore, we want to capture this additional flexibility while calculating compliance levels. In this paper, we address the issue of compliance measurement in more detail and propose new measures for (fuzzy) compliance with a clinical protocol and their interpretations. In two case studies, we analyzed two ICU protocols, namely, a glucose management protocol and a weaning protocol. The glucose management protocol defines how much insulin and/or glucose should be administered to the patients. The weaning protocol defines the step-by-step process of gradually diminishing and ultimately stopping the patients' artificial, mechanical breathing. This paper builds on and is a continuation and extension of our previous work [13,19] with the overall aim to develop methods that can evaluate protocol compliance and learn from the deviations from the clinical protocol and the patient outcomes to further improve healthcare processes.

This paper is structured as follows. The next section provides an overview of compliance measures found in the literature. Section 3 describes the newly proposed method, which is followed by the application of this method to the two case studies in Section 4: the evaluation of compliance with a glucose management protocol and a weaning protocol in the ICU of a 715-bed Dutch university medical center. Section 5 describes the qualitative evaluation of proposed measures by the intended users, and Section 6 discusses additional findings from the evaluation sessions. The paper is finished with concluding remarks.

## 2. Background and Related Work

In general, compliance aims to ensure that the operations in an organization are in alignment with the governing rules, policies, standards, regulations, and laws originating from within the organization or from external (regulatory) bodies [20]. In the last two decades, the necessity for compliance has been augmented by the increasing number of laws and regulations issued in several domains, such as finance, banking, environment, and healthcare [21]. For instance, the de Sarbanes-Oxley Act on financial record keeping and reporting for corporations was issued in 2002 [22].

With regard to the timing, organizations employ different strategies to ensure compliant operations [23,24]. The compliance assessment can take place (i) at *design-time*, that is, at the early stage of process (or operation) design; (ii) at *run-time*, that is, when the processes are running, and (iii) *ex-post* (offline), that is, after execution of the processes with audits examining the traces (logs) of executed processes. While the design- and run-time assessments are typically preventive, i.e., aim at detecting violations before they occur, ex-post assessments or audits are detective and focus on possible violations and trends [21].

The studies on compliance assurance of processes typically focus on one of the above-mentioned stages and concentrate on specific process concerns (e.g., control flow, resource, temporal, or data related) [21]. The literature reports on pattern-based approaches to aid in the formal specification of compliance rules that can be used for design-time, run-time or offline assessments (e.g., [15,25–27]). Using execution traces residing in enterprise information systems is also introduced as an alternative approach that can be employed for auditing operations in the organization [28,29]. However, despite the vast research in the process compliance research field [21], very few articles study the measurement or quantification of the compliance level. In the remainder of this section, we discuss the existing approaches, focusing particularly on those applied in the healthcare domain. In this context, we consider a clinical protocol as a domain specific process model/specification [30]. Moreover, in our work, we focus mostly on the ex-post compliance assessment.

### 2.1. Quantifying Clinical Protocol Compliance

In the medical domain, compliance metrics generally measure whether a clinical protocol was followed and to what degree. The measurement can take different perspectives, e.g., treatment compliance, patient adherence [31,32] or provider adherence [17,33] (see Figure 1). It can be measured through different methods, e.g., direct observation, self-reporting techniques, electronic monitoring, and documentation review [17].

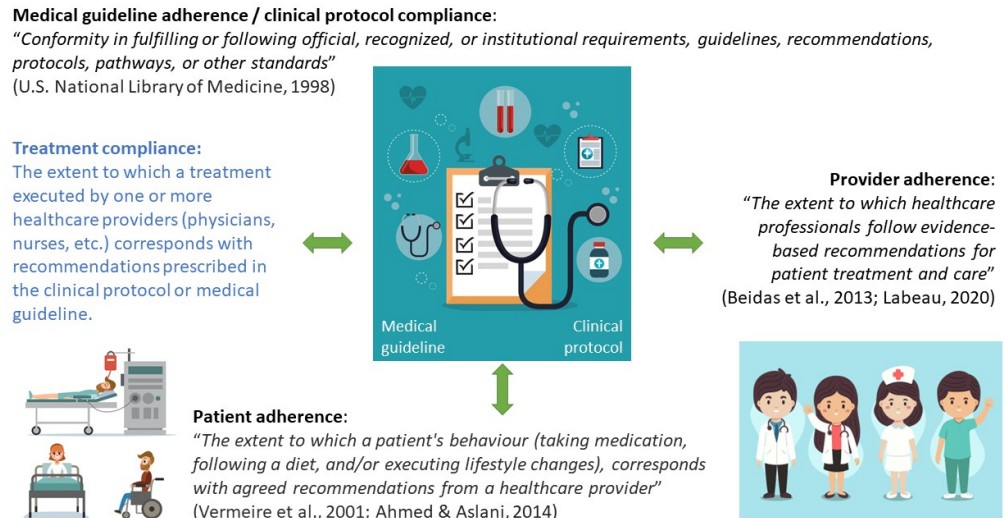

**Figure 1.** Compliance in the medical domain defined, showing the general definition by the U.S. National Library of Medicine [34] and the different perspectives of patient adherence [31,32] and provider adherence [17,33], including the perspective of treatment compliance, as proposed in this paper. (Images by Freepik).

Compliance assessment is mentioned typically in the context of *medical trials*. There, compliance is described as "adherence to all the trial-related requirements, good clinical practice (GCP) requirements, and the applicable regulatory requirements. The investigator should not implement any deviation from, or changes in, the protocol without agreement by the sponsor and prior review and documented approval/favorable opinion from the IRB/IEC (institutional review board/independent ethics committee) of an amendment,

except where necessary to eliminate an immediate hazard(s) to trial subjects or when the change(s) involves only logistical or administrative aspects of the trial (e.g., change in monitor(s), change of telephone number(s))" [35].

Another context in which protocol compliance is often mentioned is *intervention studies*. The goal of such an intervention study is to evaluate whether using the clinical protocol indeed has the desired positive effect on patient health and safety. In various intervention studies, compliance was, however, measured in quite different ways. Here we present an overview of possibilities encountered in the literature.

The protocol under consideration may define some aspects regarding the compliance measure. The first aspect concerns the *person* performing the activities from the protocol. It can be healthcare providers [12,36] or the patient or his/her family, e.g., that can evaluate the patients' compliance or adherence to the doctors' recommendations, for instance, regarding drug intake [37].

This aspect is also related to how we record the compliance. In the first case, the electronic health record (EHR) may be used to verify that certain activities were performed [38]. In the latter case, EHR information is not available, and the researchers must rely on observations [39] or human perception and judgment. In such a case, questionnaires or qualitative analysis techniques, such as focus group interviews, may be used to evaluate the compliance [40,41]. In this paper, we focus on compliance by healthcare providers.

Another important dimension resulting from the protocol is the *medical subject* of the protocol. Does the protocol concern a continuous process [42] or does it define concrete action steps [43,44]? In the first case, in [42], the authors analyze the case of continuous drug administration, namely vasopressors. The authors defined potential protocol deviations when a patient had pressure outside a predefined target range for more than four hours without adjustment of the vasopressor dose. Next, those deviations were divided into two groups as clinically-justified and non-clinically-justified. In addition to the number of deviation events, the number of days and number of patients with at least one deviation event were also reported, as well as their percentages.

In the latter case [43,44], the protocol defines concrete action steps or activities. Hence, the next logical question is how to measure compliance with those steps. Two possibilities were given in [45] by measuring whether an activity was completed within a certain, prescribed time frame or was not executed at all. Another solution was used in [36]. This work also considered the clinicians as compliant with a guideline if they stated why the guideline was not being followed.

Those individual measurements need to be aggregated as a next step and reported using either the *activity* perspective, i.e., the compliance is reported per activity [11,46], or the *patient* perspective. In this case, there are also several possibilities. For instance, reference [12] defines overall adherence to a prescribed distress screening protocol as calculated based on documentation in the EHR that screening adherence and an appropriate clinical response had occurred. Only when both the screening and response took place, overall adherence per patient was considered "yes". The authors calculated the overall adherence rate as the number of overall adherent patients per number of all patients. Some authors, such as Stewart et al. [43], introduced a scoring system. Each protocol execution could receive a score between 0 and 5, where score 4 or 5 allowed one deviation from the protocol and meant excellent protocol execution. The adherence was reflected by the percentages of the protocol executions with each of the six scores.

Another approach was used in [38]. The authors defined a number of indicators operationalizing the guidelines' requirements that served as measures of guideline adherence. The presence of an indicator was given a score, resulting in a point score for a protocol. Similarly, Drews et al. [44] calculated the number of steps that were compliant with reporting the percentages of executions with zero, one, or two non-compliant activities. Other authors decided to calculate a non-compliance rating (NCR) as the number of recommendations not acted upon [47].

To summarize, in the current literature, the following aspects are considered in the assessment of compliance, in particular in the healthcare context:

- Measurement based on: objective data (from hospital information systems) vs. subjective data (from questionnaires),
- Main actor: health provider vs. patient (or his family),
- Perspective: protocol level vs. activity/action level,
- Medical subject: continuous protocol vs. protocols with steps,
- Approach to deviations: deviations allowed after explanation vs. no deviations allowed.

### 2.2. Compliance Checking Using Process Mining

In recent years, process mining has become increasingly popular as a technique for offline compliance checking, also called auditing [21,48,49]. Process mining is an analysis technique that uses process execution logs or traces that reside in an organization's information systems [50]. In general, process mining can be used (i) for *discovering* process models from the execution logs, (ii) for process model *enhancement*, and (iii) for *conformance checking* [51]. Conformance checking enables the organization to assess whether the processes (operations) were correctly executed with respect to the prescribed set of rules [28].

Process mining is also applied to the healthcare domain [51–58], but the focus has mainly been on discovering a process model from the event log of diagnosis and treatment activities, often leading to 'spaghetti models'. The goal of conformance checking for healthcare processes has received less attention thus far [59–61].

Conformance checking assumes the existence of a process model and the observed behavior of actors as they execute the process. This observed behavior is recorded in an execution log, which consists of a list of the tasks that were performed, their timestamps and identifiers of the case for which the task was performed. Additional information, such as the actor performing the task and information about the task, can enable further analysis [62,63]. In general [62–64], four metrics are proposed that aim to quantify the level of compliance:

- Fitness indicates the degree to which log traces (in our context protocol executions) can be linked with valid execution paths in the process model, i.e., can the model reproduce the log traces?
- Appropriateness refers to the degree of accuracy to which the process model describes the observed behavior.
- Precision indicates how well a model represents the behavior seen in a log, especially in the context of superfluous activities.
- Generalization indicates whether the process model represents the process paths in a generic way or only the behavior observed in the data.

In the healthcare context, a clinical protocol can be considered a process model and the information collected in the EHR can (partially) represent the observed behavior as an execution event log [30]. Hence, the fitness refers to compliance with the protocol, while the other metrics can be considered indicators for protocol validity, which is beyond the scope of this paper.

In order to calculate the fitness, one first needs to align or replay the captured behavior in the form of execution traces with the (predefined) process model. This alignment will result in calculating four values, namely $p$–the number of produced tokens, $c$–the number of consumed tokens, $m$–the number of missing tokens and $r$–the number of remaining tokens. Ideal alignment results in $m = 0$ and $r = 0$. The fitness of a single trace is calculated as

$$fitness(\sigma, N) = \frac{1}{2}\left(1 - \frac{m}{c}\right) + \frac{1}{2}\left(1 - \frac{r}{p}\right) \tag{1}$$

where $N$ is the process model and $\sigma$ is the trace. Fitness can be interpreted as a distance between the log traces and the closest valid execution path, and hence, assess process compliance. More details can be found in [50,64].

### 2.3. First Approach Towards Fuzzy Compliance

In the work shown above, an activity could be only compliant or non-compliant with the protocol. In [13], we propose to consider compliance of a single activity not as a binary state, but as a matter of degree. In this approach, each activity in a protocol is associated with a compliance level. For instance, consider an activity of drug administration. There are several dimensions in which it can be compliant with the protocol: the order, or the place in which it happened in the execution trace (cf. process mining), the timing (did it happen in the right time frame after, e.g., the previous activity), the dose, etc. For this illustrative case, let us consider the dose dimension only. In this case, the compliance level can be defined using a fuzzy set with a trapezoidal membership function, e.g., as shown in Figure 2, with four clearly defined threshold points in which dose deviation is perfectly OK (compliance degree of 1) and dose deviation is completely not OK (compliance degree of 0). For the values in-between, we use linear interpolation. The use of such membership function was already advocated by Zadeh [65] since it is easy to understand by domain experts and provides a good compromise between a so-called cointension and computational complexity. In this context, cointension indicates the degree between the explicit semantics, defined by the formal parameter settings of the model and the implicit semantics conveyed to the reader by the linguistic representation of knowledge.

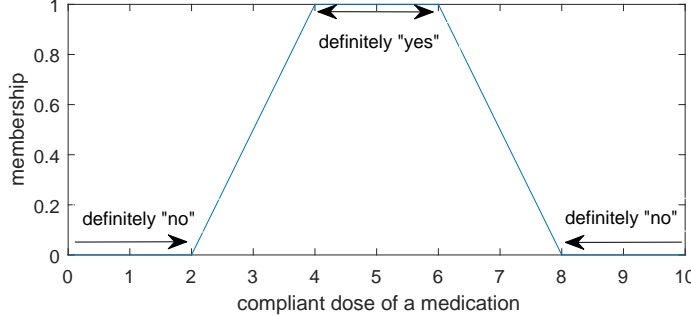

**Figure 2.** Membership of a fuzzy set defining compliant dose of a medication.

In [13], the compliance degree of the protocol refers to the average compliance levels of all the activities performed within this protocol for all patients. In this paper, we propose to extend the fuzzy compliance metric proposed in [13] by taking into consideration the patient (trace) and importance of the activities.

### 3. Proposed Metrics of Protocol Compliance

We will use the following notation in this paper. Let us consider a protocol $P$, which consists of $n$ unique activities $p_i$, $i = 1, \ldots, n$. The protocol $P$ may include decision points, as well as loops in which certain activities can be repeated. For instance, consider a protocol shown in Figure 3. In this protocol, there are three unique activities $A$, $B$ and $C$. Therefore, $P = \{A, B, C\}$. First, activity $A$ should be performed. Next, activity $B$ is iteratively performed until value $x$ exceeds a certain threshold value Theta.

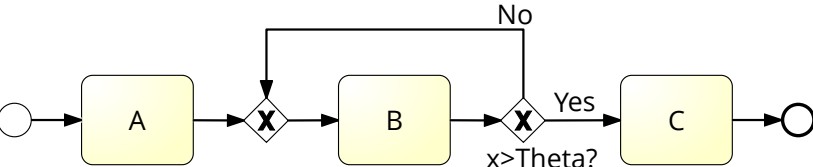

**Figure 3.** Example of a clinical protocol.

The behavior of healthcare providers is captured in the form of traces $T$. Let us assume that the protocol was followed for $m$ patients. Hence, there are $m$ executions, traces of this protocol, $T = \{t_1, t_2, \ldots, t_m\}$. A trace $t_j$ contains information about $r_j$ activities performed related to this protocol. For instance $t_1 = \langle A, B, B, C \rangle$ and $t_2 = \langle A, B, C \rangle$.

Therefore, for each activity $p_i$, the healthcare providers may define rules defining the degree of compliance. This compliance degree $\mu_{p_i}(\cdot)$ of an activity $p_i$ is defined as the membership value of a fuzzy set [13]. The pseudo-code of the algorithm is depicted in Algorithm 1.

---

**Algorithm 1** Protocol compliance

---

for each patient in the protocol
    find the best alignment with the protocol
    for each activity
        calculate the difference in the properties between the activity performed and protocol
        calculate compliance degree
aggregate compliance degrees over all activities and all patients (c.f. Algorithms 2–4)

---

*3.1. Compliance per Patient*

In [13], all the memberships of all the activities of all patients are averaged, ignoring the information about which activities were performed for which patient. Moreover, averaging may not necessarily be the best aggregating operator. In some cases, one may suggest that the protocol is compliant with the degree to which the least compliant activity is compliant or most of the activities are compliant.

Therefore, there can be a need to first estimate the compliance level per patient and, in the second step, aggregate those compliance levels. We suggest using the ordered weighted averaging (OWA) operator [66,67] for this purpose.

For completeness, let us first repeat the definition of an OWA operator. An ordered weighted averaging (OWA) operator of dimension $n$ is a mapping

$$F : [0, 1]^n \longrightarrow [0, 1] \tag{2}$$

if associated with $F$ is a weighting vector

$$W = [w_1, \ldots, w_n]^T \tag{3}$$

such that:

1. $w_i \in [0, 1]$, for all $i = 1, \ldots, n$,
2. $\sum_{i=1}^{n} w_i = 1$, and

$$F(a_1, \ldots, a_n) = W^T B = \sum_{j=1}^{n} w_j b_j \tag{4}$$

where $b_j$ is the $j$-th largest element in the set $\{a_1, \ldots, a_n\}$, and $B = [b_1, \ldots, b_n]$. $B$ is called an ordered argument vector if for each $b_i \in [0, 1]$, $j > i$ implies $b_i \geq b_j$, $i = 1, \ldots, n$.

An equivalent notation is one with increasing sorting.

The minimum can correspond to the universal quantifier *for all*, while the maximum can correspond to the existential quantifier *for at least one*. Moreover, by an appropriate choice of the weighting vector $W$, between $W = [1, 0, \ldots, 0]$, as in the maximum type aggregation, and $W = [0, \ldots, 0, 1]$, as in the minimum type aggregation, we can obtain an aggregation operator corresponding to "intermediate" *linguistic quantifiers*, e.g., *at least a half, most, almost all*, etc. For regular non-decreasing monotone quantifiers, Yager [66] generates the weighting vector $W = [w_1, \ldots, w_n]^T$ as:

$$w_i = \mu_Q\left(\frac{i}{n}\right) - \mu_Q\left(\frac{(i-1)}{n}\right), \qquad i = 1, \ldots, n \tag{5}$$

and since, by definition, $\mu_Q(0) = 0$ and $\mu_Q(1) = 1$, then $w_1 + \cdots + w_n = 1$. Therefore, an OWA operator may be viewed to provide, in general, a linguistic quantifier driven aggregation [67]. Please note that an averaging can also be performed with an OWA operator with the weighting vector $W = [\frac{1}{n}, \ldots, \frac{1}{n}]$.

In our context, first we calculate the compliance level for each trace (patient) $t_j$, $j = 1, \ldots, m$ as

$$c_{OWA}(t_j) = F(\mu_p(a_{j1}), \ldots, \mu_p(a_{jr_j})) = \sum_{k=1}^{r_j} w_k b_k \tag{6}$$

where $r_j$ is the number of activities in the trace $t_j$, and $b_k$ is the $k$-th largest element in the set $\{\mu_p(a_{j1}), \ldots, \mu_p(a_{jr_j})\}$, i.e., set of compliance degrees of individual activities in trace $t_j$. $w_k$ are the weights calculated according to (5). The quantifier can be defined linguistically, such as *for all*, *most*, *almost all*, depending on the user's preference. This quantifier enables setting a fuzzy threshold to what percentage of patient population is expected to follow the protocol and to what extent. For instance, in cancer treatment protocols, around 80% of patients follow the protocols.

Next, we aggregate those compliance levels of the patients. For this purpose, we can also use an OWA operator.

$$C_{OWA}(P) = F(c_{OWA}(t_1), \ldots, c_{OWA}(t_m)) \tag{7}$$

Again, the quantifier can be defined linguistically, such as *for all*, *most*, *almost all*, depending on the user's preference, or one can use an average. In such case, we can interpret the obtained value. For instance, if we use a linguistic quantifier *most* to calculate the weighting vectors, the obtained value can be interpreted as the degree to which most patients have most of the activities compliant with the protocol. The pseudo-code for calculating OWA-based compliance is listed in Algorithm 2. The use of quantifiers enables healthcare providers to include an expectation regarding compliance or non-compliance in the patient population.

---

**Algorithm 2** OWA-based compliance per patient

---

define targeted overall compliance
define targeted compliance of individual patient
for each patient in the protocol
   calculate patient's compliance using (6)
calculate overall compliance using (7)

---

### 3.2. The Criticality Level of Activities

The clinical protocol typically incorporates a number of activities, each with potentially different levels of criticality (relative importance) to the desired outcome of the protocol. For instance, compare "administering the correct dosage for an essential drug for a specific type of patient" with "checking the pillow/patient position". In cases where a protocol incorporates activities of different relative criticality, the compliance level of a relatively more critical activity should have a larger influence on the overall level of compliance. Therefore, if available, this information should also be incorporated into the compliance metric.

For this purpose, we propose using the OWA operator with an importance qualification [67,68]. The importance qualification ($I$) transforms the weighting vector of the OWA operator from $W$ into $\overline{W}_I$. The weights of the weighting vector $\overline{W}$ are defined as

$$\overline{w}_j = \mu_Q\left(\frac{\sum_{k=1}^{i} u_k}{\sum_{k=1}^{n} u_k}\right) - \mu_Q\left(\frac{\sum_{k=1}^{i-1} u_k}{\sum_{k=1}^{n} u_k}\right) \tag{8}$$

where $u_j$ is the importance of the $j$-th largest element of the set $\{a_1, \ldots, a_n\}$ [68].

In our context, first we calculate the compliance level for each trace (patient) $t_j$, $j = 1, \ldots, m$ as

$$c_{OWA_I}(t_j) = F_I^W(\mu_p(a_{j1}), \ldots, \mu_p(a_{jr_j})) = \sum_{k=1}^{r_j} \overline{w}_k b_k \qquad (9)$$

where $r_j$ is the number of activities in the trace $t_j$, and $b_k$ is the $k$-th largest element in the set $\{\mu_p(a_{j1}), \ldots, \mu_p(a_{jr_j})\}$, i.e., set of compliance degrees of individual activities in trace $t_j$. $\overline{w}_k$ are the weights calculated according to (8), also taking into consideration the level of criticality or importance. Depending on the user's preference, the quantifier can also be defined linguistically with, *for all*, *most*, *almost all*.

Next, we aggregate the compliance levels of the patients. We assume, however, that all patients are equally important. For this purpose, we use an OWA operator.

$$C_{OWA_I}(P) = F(c_{OWA_I}(t_1), \ldots, c_{OWA_I}(t_m)) \qquad (10)$$

Depending on the user's preference, averages can be used or the linguistic quantifiers (e.g., *all*, *most*, *almost all*, etc.) can be defined. We can interpret the obtained value depending on the used quantifiers, e.g., as the degree to which most patients have most of the important activities compliant with the protocol, provided that we use the linguistic quantifier *most* to calculate the weighting vectors. The pseudo-code for calculating OWA-based compliance with the criticality of the activities is listed in Algorithm 3. The benefits of this method are that the use of quantifiers includes an expectation regarding compliance or non-compliance in the patient population and includes the importance factor for the activities.

---

**Algorithm 3** OWA-based compliance with criticality of the activities

---

define targeted overall compliance
define targeted compliance of an individual patient
establish importance of the individual activities
for each patient in the protocol
    calculate weights using (8)
    calculate patient's compliance using (9)
calculate overall compliance using (10)

---

### 3.3. Sugeno Integral-Based Compliance Measure

The difficulty with the OWA-based method is the weighting vector. Even though the weights can be calculated from a linguistic quantifier, several possibilities may exist. This can be considered a drawback of those methods. In addition, the obtained numbers might be relatively difficult to interpret (e.g., low values in these cases can be considered less informative.) Therefore, aggregation using a Sugeno integral offers an interesting solution.

First, let us reiterate the definitions of fuzzy measure and Sugeno integral. Let $X = \{x_1, x_2, x_3, \ldots, x_n\}$ be a finite set. A fuzzy measure [69,70] on $X$ is a function $g : 2^X \to [0, 1]$ such that:

- $g(\varnothing) = 0$
- $g(X) = 1$
- If $A \subseteq B$ then $g(A) \leq g(B) \ \forall A, B \in 2^X$

Moreover, there is a special case of fuzzy measure called a $\lambda$-fuzzy measure or Sugeno $\lambda$-fuzzy measure [71]. The $\lambda$-fuzzy measure is a fuzzy measure $g$ that satisfies the $\lambda$-rule, i.e., there exists $\lambda \in (-1, \infty)$ such that $g(E \cup F) = g(E) + g(F) + \lambda g(E) \cdot g(F)$ whenever $E \cap F \neq \varnothing$.

When $g(X) = \sum_{i=1}^{n} g(\{x_i\})$, the parameter $\lambda = 0$; we then obtain probability measure. The values $\{g(\{x_i\}) | i = 1, 2, \ldots, n\}$ are called the fuzzy densities. Knowing the densities, parameter $\lambda$ can be uniquely determined by the equation [71]:

$$g(X) = \frac{1}{\lambda} \left[ \prod_{i=1}^{n} (1 + \lambda \cdot g(\{x_i\})) - 1 \right] \tag{11}$$

This equation comes directly from the $\lambda$-rule and the fact that $X = \bigcup_{i=1}^{n} \{x_i\}$. In our work, we use $\lambda$-fuzzy measures.

Let $g$ be a fuzzy measure and $h$ be a function $h : X \rightarrow [0,1]$. Moreover, assume that $\{x_i\}$ are ordered so that $h(x_1) \geq h(x_2) \geq \ldots \geq h(x_n)$. Then, the discrete Sugeno integral [71] of a function $h$ with respect to $g$ is a function $S_g : [0,1]^n \rightarrow [0,1]$ such that

$$S_g(h) = \max_{i=1,\ldots,n} [\min(h(x_i), g(A_i))] \tag{12}$$

where $A_i = \{x_1, x_2, \ldots, x_i\}$.

In our context of protocol compliance, the Sugeno integral can have a very interesting interpretation. Let us consider a single protocol execution. The value of k% for the Sugeno integral would mean that at least k% of activities are compliant with at least k% degree. On the protocol level, we could say that at least k% traces are compliant with the k% degree.

As discussed previously, we first calculate the compliance level for a trace $t_j$, $j = 1, \ldots, m$. Trace $t_j$ had $r_j$ activities $a_{kj}$, $k = 1, \ldots, r_j$. In our context, set $X$ is the set of activities performed in the trace $t_j$, so $X = \{a_{1j}, \ldots a_{r_j j}\}$. We consider two possibilities for the densities. In the first case, all activities are equally important and a density of an activity $a_{kj}$ is

$$g_1(a_{kj}) = \frac{1}{r_j} \tag{13}$$

In this case, the densities add up to one, hence $\lambda = 0$, and we use a probability measure.

In the other case, we incorporate the importance $u_{kj}$ of the activity $a_{kj}$. We assume that the importance is a number from interval $[0,1]$, where 1 denotes the highest and zero denotes the lowest level of importance/criticality. Here, the density of an activity $a_{kj}$ is

$$g_1(a_{kj}) = \frac{u_{kj}}{r_j} \tag{14}$$

In this case, several densities will be less than $\frac{1}{r_j}$, and therefore, the sum will be less than 1, producing a $\lambda$ that is bigger than 0. Intuitively, activities that are more important have a larger measure. They also have more influence on the compliance level. Certainly, other choices of densities can be followed.

We define the compliance for a trace as the Sugeno integral of the function $f(x) = x$ with respect to measure $g_1$.

$$c_S(t_j) = \max_{\alpha \in [0,1]} \left( \min(f(\alpha), g_1(t_{j\alpha})) \right) \tag{15}$$

where $t_{j\alpha} = \{a_{kj} : k = 1, \ldots, r_j \text{ and } \mu(a_{kj}) > \alpha\}$, i.e., a set of activities performed in the trace $t_j$ that are compliant at least with the level $\alpha$.

Once the compliance levels for the traces are calculated, we can aggregate those values using the Sugeno integral again. Here, $X = t_1, \ldots, t_m$. Since each trace (patient) is equally important, we use a probability measure and the densities are

$$g_2(t_j) = \frac{1}{m} \tag{16}$$

We define the compliance for a protocol $P$ as the Sugeno integral of function $f(x) = x$, with the measure $g_2$.

$$C(P) = \max_{\alpha \in [0,1]} \left( \min(f(\alpha), g_2(T_\alpha)) \right) \tag{17}$$

where $T_\alpha = \{t_j : j = 1, \ldots, m \text{ and } c_S(t_j)) > \alpha\}$, i.e., a set of traces $t_j$ that are compliant with at least the $\alpha$ level. The pseudo-code for calculating the Sugeno integral-based compliance measure is listed in Algorithm 4.

---

**Algorithm 4** Sugeno integral-based compliance measure

---

establish the importance of individual activities, if needed
for each patient in the protocol
　　calculate densities as (13) or (14)
　　find $\lambda$ using (11)
　　calculate compliance per patient using (15)
calculate densities as (16)
calculate overall compliance using (17)

---

The advantage of this method is that it combines activity compliance, patient compliance and protocol compliance as a single number. This number could be used as a KPI, and the calculations behind it could help identify the biggest discrepancies at the patient and activity levels.

## 4. Case Studies

To demonstrate the compliance of the proposed methods, we used two protocols: the glucose management protocol and the weaning protocol of the intensive care unit (ICU) as currently used in the Maastricht University Medical Centre+, Maastricht, The Netherlands. The Maastricht University Medical Center ICU consists of three different wards. Two of them, denoted as *A* and *B*, treat mixed medical/surgical patients, where most patients (90%) are acutely ill patients (requiring immediate assessment or treatment). Ward *C* treats mostly elective cardiac patients after planned surgery.

The purpose of the glucose management protocol is to control and stabilize the patients' glucose levels so that it remains within safe intervals. Hyperglycemia (glucose level above 11.1 mmol/L) is common in critically ill patients, both with and without diabetes, e.g., due to the stress factor [72–74]. Both hyperglycemia and hypoglycemia (glucose level below 2.2 mmol/L) are associated with a higher risk of mortality and morbidity [75,76].

The weaning protocol describes the process of decreasing the level of ventilator support, including extubation (removal of the tube from the patient's airway). The aim of the protocol is to safely reduce the level of ventilator assistance as fast as possible because of the patients' risk increase with prolonged duration mechanical ventilation. This protocol is used for postoperative cardiothoracic surgery patients in the ICU in ward *C*.

### 4.1. Glucose Management Protocol Case Study

The glucose management protocol is a nurse-driven protocol with the aim of controlling and maintaining stable glucose levels in patients. The process is relatively simple; a nurse measures the glucose value of a patient either using point-of-care measurement or central laboratory determination using arterial blood samples. Based on the result, he/she adjusts the settings of the perfusor (i.e., machine that administers drug to the patient on a continuous basis).

In this paper, we study the wards A and B, where the same glucose management protocol (shown in Tables 1 and 2) has been implemented for many years. Therefore, in this paper, we analyze wards *A* and *B*. The protocol under consideration concerns intravenous administration of short-acting insulin and applies to every patient till the moment when she/he can receive oral nutrition. The protocol assumes that the target values of blood glucose are between 4.5 and 7.0 mmol/L. The starting scheme (perfusor settings for a new patient) is shown in Table 1, and the adaptation schemes for the perfusor settings are presented in Table 2.

**Table 1.** Starting scheme of the glucose management protocol used in the study.

| | Starting Schema | |
|---|---|---|
| Blood Glucose (mmol/L) | Insulin Perfusor Setting (50 units/50 mL NaCl 0.9%) | Insulin Bolus |
| <7.0 | - | |
| 7.0–8.0 | 1 unit/h | |
| 8.0–10.0 | 2 unit/h | |
| 10.0–15.0 | 4 unit/h | 4 units insulin |
| 15.0–20.0 | 6 unit/h | 6 units insulin |
| >20.0 | 6 unit/h | 8 units insulin |

**Table 2.** Perfusor adaptation scheme for glucose management protocol used in the study.

| | Blood Glucose Decreased > 30% or Increased | |
|---|---|---|
| Blood Glucose (mmol/L) | Insulin Perfusor Setting (50 units/50 mL NaCl 0.9%) | Action |
| <3.5 | stop | act according hypoglycemia protocol |
| 3.5–4.5 | −1.0 unit/h | check glucose after 30 min |
| 4.5–7.0 | no changes | |
| 7.0–8.8 | +0.5 unit/h | |
| 8.0–9.0 | +1.0 unit/h | |
| 9.0–10 | +1.5 unit/h | |
| 10–15 | +2.0 unit/h | 2 units insulin as bolus |
| >15 | +3.0 unit/h | 4 units insulin as bolus |
| | Blood Glucose Decreased < 30% | |
| Blood Glucose (mmol/L) | Insulin Perfusor Setting (50 units/50 mL NaCl 0.9%) | Action |
| <3.5 | stop | act according hypoglycemia protocol |
| 3.5–4.5 | stop, if glucose > 5 mmol/L then start with half of the last dose | check glucose every 15 min till glucose > 5 mmol/L |
| 4.5–7 | half the dosage | |
| 7–10 | −1 unit/h | |
| >10 | no changes | |
| <3.5 | stop | 50 mL glucose 50% in 10 min and check glucose level |
| >4.5 (after 1st glucose bolus) | start with last setting −1 unit/h | check glucose after 30 min |
| <4.5 (after 1st glucose bolus) | keep perfusor stopped | 30 mL glucise 50% in 10 min and check glucose level |
| >4.5 (after 2st glucose bolus) | start with last setting −2 unit/h | |
| <4.5 (after 2st glucose bolus) | keep perfusor stopped | 50 mL glucose 50% in 10 min and consult the doctor |

In our study, we measured the perceived compliance measure of the wards' coordinators and nurses working on those two wards and calculated compliance levels according to different metrics. First, we interviewed the wards' coordinators and nurses regarding the awareness, use and reasons for (non)adherence to this protocol. In ward *A*, the interviewees indicated that they did not follow the protocol (anymore), although many of them (57%) were aware of it. Controlling and managing the blood glucose level is achieved based on the experience and insight of nurses.

In ward *B*, the protocol was still in use, and its physical (paper) version was available in the workplace. All nurses confirmed that they were aware of the protocol and most of them were following the protocol to a large extent.

We received data from those two wards *A* and *B*, containing the glucose-management-related activities from years 2014 and 2015. The data included the blood glucose results and changes in the insulin perfusor settings as well as the information regarding insulin and glucose bolus (injections). There were 297 patients in Ward *A* and 274 patients in Ward *B*. The frequency of glucose measurement activity was not stated in the protocol (except for the hypoglycemia case); therefore, we also assessed the compliance regarding the medication dosage, i.e., insulin perfusor settings, insulin bolus and glucose bolus.

In order to measure fuzzy compliance levels, the healthcare providers defined clinically accepted and non-accepted deviations by proving two threshold values (completely acceptable deviation and completely unacceptable deviation) for each of the actions (insulin perfusor settings, insulin bolus and glucose bolus) so that a trapezoidal membership function could be defined. Those values are presented in Table 3. The degree of compliance with an activity was defined as the minimal value of compliance with respect to the three medication types.

The results of different compliance measures are shown in Table 4. The very low values of the strict compliance for Ward *B* were surprising in comparison to the results of the interview, as both the coordinator and nurses believed that they were following the protocol. The fuzzy compliance results are naturally much higher. The fuzzy compliance (average based) [13] of 83% indicated that not all activities are within the acceptable deviation threshold. The newly proposed methods look at the compliance, also taking into account "per patient" compliance. In the example, we have used three quantifiers: many (Trap[0.5, 0.7, 1, 1]), most (Trap[0.6, 0.8, 1, 1]), and almost all (Trap[0.8, 0.9, 1, 1]).

**Table 3.** Thresholds for accepted and non-accepted deviations.

| Medication | Totally Acceptable Deviation | Totally Unacceptable Deviation |
|:---:|:---:|:---:|
| Perfusor setting | 1 unit/h | 2 units/h |
| Insulin bolus | 2 units | 3 units |
| Glucose bolus | 10 mL | 15 mL |

**Table 4.** Compliance values for Wards *A* and *B* according to different metrics.

| Method | Ward *A* | Ward *B* |
|:---:|:---:|:---:|
| Strict Compliance | | |
| % of compliant executions | 1.7% | 0.7% |
| % of compliant activities | 37.8% | 34.8% |
| Fuzzy compliance | | |
| Fuzzy compliance (average based) [13] | 82.6% | 82.8% |
| OWA-based fuzzy compliance | | |
| most ($Q_1$) patients have most ($Q_2$) of their activities compliant | 62.1% | 75.8% |
| many ($Q_1$) patients have almost all ($Q_2$) of their activities compliant | 23.6% | 26.6% |
| Sugeno integral-based fuzzy compliance | | |
| Sugeno integral-based | 70.4% | 71.4% |

We considered two scenarios here: one, where we assume that most patients follow the protocol well (most activities compliant), which corresponds to a linguistic summary "most ($Q_1$) patients have most ($Q_2$) of their activities compliant", and the second one where many patients follow the protocol very well (almost all activities compliant), which corresponds to a linguistic summary "many ($Q_1$) patients have almost all ($Q_2$) of their activities compliant".

In the first case, Ward *B* has higher values, indicating that in most cases they are more compliant than staff in Ward *A*. In the case of the second summary, both wards do not score well, indicating that overall long-term compliance per patient is hard to obtain. Naturally, with different definitions of quantifiers, different truth values of the summaries could be obtained.

The Sugeno integral-based fuzzy compliance also offers an interesting perspective. The obtained value can be interpreted as follows: at least 70% of patients have 70% of activities compliant to a degree of 70%. The advantage is that with one number, we can capture the overall compliance level.

### 4.2. Weaning Protocol Case Study

In Ward *C*, we analyzed the weaning protocol for patients that were ventilated for less than 72 h. During a surgical procedure, the ventilation of the patient is taken over by a mechanical ventilator. After surgery, the ventilator continues to support the patient's ventilation until sufficient spontaneous breathing activity occurs. The process of reduction in the level of support is a stepwise procedure, described in the protocol and shown in Figure 4.

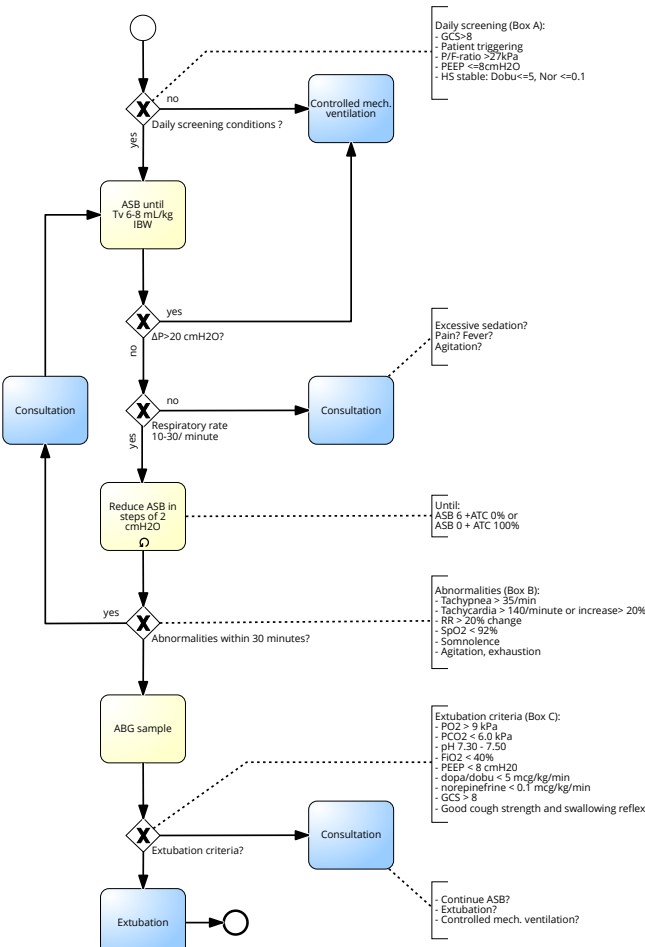

**Figure 4.** The weaning protocol used in the case study.

The weaning protocol can be divided into three basic phases: Bi-level positive airway pressure (BIPAP), assisted spontaneous breathing (ASB) and extubation phases (stopping support by mechanical ventilation). In the BIPAP phase, the whole breathing is taken over by the mechanical ventilator. When a patient fulfills the criteria from Note A, he/she can enter the ASB Phase, in which the breath is initiated by the patient and supported by the ventilator. The support of the mechanical ventilator is gradually decreased, while the state

of the patient is monitored by the nurses. When the patient is fully breathing without any support from the ventilator and is stable for 30 min (as described by note B), he/she enters the extubation phase. The patient is extubated when he/she fulfills the criteria in Note C, which means that the tube is removed and the patient is disconnected from the mechanical ventilator. This ends the weaning protocol.

We received the data of 2473 patients that underwent cardiothoracic surgery between August 2012 and April 2015. The data contained information regarding patient-related indicators, measured with different frequency, either automatically or manually. In addition, the settings of the ASB machine were captured, unfortunately only once in every 30 min, and not with each change in the values. Some variables or activities were not registered.

Therefore, for this protocol, with the available data, one possibility is to focus on the compliance at the decision points of whether to start decreasing the ventilatory support while breathing and whether to extubate the patient. We could detect those points for 1680 patients. The decisions are guided by several conditions described by Note A and Note C, respectively. The thresholds present in Notes A and C, for which we received the data, are shown in Table 5. This table also shows the deviation thresholds and importance of each of those criteria. For note A, we check six criteria, and for note C, we check eight criteria.

**Table 5.** Note A + Note C thresholds.

| Condition | Compliance mf | Importance |
|---|---|---|
| note A | | |
| GCS > 8 | Trap[8,10,∞, ∞] | very important |
| $PaO_2/FiO_2$ ratio > 27 | Trap[23, 28, ∞, ∞] | very important |
| PEEP ≤ 8 | Trap[0, 0, 8, 12] | important |
| $FiO_2$ < 50% (Note A) | Trap[0, 0, 40, 65] | important |
| note C | | |
| $FiO_2$ ≤ 40% (note C) | Trap[0, 0, 40, 55] | very important |
| $PO_2$ ≥ 9 kPa | Trap[8, 9. 5,∞, ∞] | very important |
| $PCO_2$ ≤ 6.0 kPa | Trap[0, 0, 5.8, 7] | important |
| pH 7.3–7.5 | Trap[7.23, 7.35, 7.45, 7.6] | very important |
| Dobutamine ≤ 5 | Trap[7.5, 15, ∞, ∞] | important |
| Norephinephrine ≤ 0.1 | Trap[0.05, 0.25 ∞, ∞] | important |

Table 6 shows different compliance values according to different measures. First, there were many missing data—for note A around 44% and for note C around 30%. Therefore, we decided to treat the unknown information as non-compliant. This is a natural and straightforward interpretation, as there is no evidence that the protocol was being followed in those cases. This vast amount of missing data causes all the compliance values to be low, especially for measures with higher compliance expectations (i.e., most patients have most of their activities compliant). We use this approach since the protocols are created from the evidence of past cases that reflect, e.g., the best known, used and applicable conditions that influence the process, as in the case of the weaning protocol. Moreover, we can also see the influence of the activity importance on the compliance values. This is especially visible when we calculate the compliance for the note C condition, where almost 15% are strictly non-compliant. The fuzzy compliance enables assessing whether the violation of the strict threshold was substantial or only marginal.

**Table 6.** Compliance values for different compliance measures.

|  | Note A | Note C | All |
|---|---|---|---|
| Strict compliance | | | |
| % of compliant conditions | 53.4% | 55.5% | 54.5% |
| % of fully compliant checks | 0.12% (2) | 0.06% (1) | 0 |
| % of non-compliant conditions | 2.8% | 14.9% | 9.7% |
| Fuzzy compliance | | | |
| Fuzzy compliance (average based) [13] | 54.2% | 43.9% | 48.3 % |
| OWA-based fuzzy compliance | | | |
| most ($Q_1$) patients have most ($Q_2$) of their activities compliant | 0 | 0 | 0 |
| some ($Q_1$) patients have most ($Q_2$) of their activities compliant | 66.7% | 24.4% | 41.8% |
| OWA-based fuzzy compliance with activity importance | | | |
| most ($Q_1$) patients have most ($Q_2$) of their important activities compliant | 0 | 0 | 0 |
| some ($Q_1$) patients have most ($Q_2$) of their important activities compliant | 66.7% | 36.8% | 47.9% |
| Sugeno integral-based fuzzy compliance | | | |
| Sugeno integral-based | 50% | 50% | 50% |
| Sugeno based with activity importance | 48.2% | 42.2% | 41.5% |

## 5. Evaluation of the Proposed Measures

For expert validation of the results of the developed analysis techniques, we interviewed two ICU-doctors and two ICU-nurses (from MUMC+) on the understandability and the usefulness of the linguistic summaries. Below, a summary of our findings is presented as well as a concluding discussion.

We used semi-structured interviews with nine questions regarding the protocol compliance measures developed as a research tool. The questions first explored the meaning and importance of protocol compliance, then how understandable some example sentences from the linguistic summaries of their ICU data were, and finally, how useful they think the linguistic summaries would be in their daily practice. The usefulness and understandability (as a component of ease-of-use) are two latent variables that have the most influence on the adoption of a new technology, including tools, techniques, methods and guidelines [77,78]. The questions are presented in Appendix A. We recorded the interviews, transcribed them, and next, we analyzed them thematically.

During the interviews, it was acknowledged by all interviewees that protocol compliance is very important: *"Yes, I personally think that it [compliance to clinical protocols and guidelines in ICU] is very important. For me as well as a newly qualified nurse. This is just this one thing that gives you the guidance how to use certain equipment or how to make some medicine or how to give some medication. I use them a lot."* (Nurse C).

**Understandability.** When being presented with a set of linguistic summaries, all interviewees agreed that they are understandable, although the nurses had some concerns with the current way of working: *"My first response would be, they [almost all patients have some of their activities compliant] all are understandable. Whether I find them useful, I think that the first two are useful to me, and maybe the third one is less useful in practice, although you could argue that almost all patients have only some of their activities compliant, so that there is a poor compliance to the protocol. They are all understandable, and relevant".* (Doctor W) *"I understand the sentences well. For instance the first one "many patients have almost all of their activities compliant" is less strong then most patients. Almost all patients, means that there are almost no patients non-compliant. For me it is understandable."* (Nurse S) *"Yes, they [some patients have most of their activities compliant] are clear. Of course, we as ICU we, mostly work with numbers, percentages, so we are not used to work with this kind of information."* (Nurse S) *"Yes, it is comprehensible. [... but] I'm dyslectic. They all look the same."* (Nurse C).

**Usefulness.** When discussing the usefulness of the linguistic summaries, we identified different points of view between the doctors and the nurses. The doctors see an added value of linguistic summaries in communication with and training of nurses: "*We [doctors] are really used to work with percentages, and numbers. And we think this gives more insight, especially for the nurses, because glucose protocol is a nurse driven protocol, and to make the effect of the compliance more insightful is good to use these sentences, it gives more insight than just saying that 55% of patients are compliant. So I think it is useful.*" (Doctor D) "*At the bedside, so if you are discussing at the bedside, with the other doctors, with the doctors in training, with the nurses, then this is more useful than using the percentages* [. . . ] "*I think [linguistic summaries are useful] at the bedside, in direct patient care, and also for educational purposes, so for education and training.*" (Doctor D) *"You could also argue that this is useful in the inter-professional collaboration, because obviously the quantitative measurements and the percentages can be relatively easily interpreted by the doctors, but it is more difficult for the nurses. So when collaborating inter-professionally on this particular topic, looking at compliance to protocols, the linguistic summaries are far more understandable for nurses, who are not academically trained like we as doctors. So this could be an advantage. And obviously this means that when you feedback the results of the analysis to the nurses, this is far more understandable, so perhaps this is also what Doctor D thinks with education and training. If you want to provide information on protocol compliance, linguistic summaries are the things to use with nurses, and also perhaps with nurse assistants and physician assistants."* (Doctor W).

However, the (experienced) nurses currently doubt whether the linguistic summaries are helpful for them since they are already trained to work with the hard numbers: *"The first sentence is the one that jumps out. So many is a lot, but actually how many. And almost all activities sounds good. And compare this to some of their activities, and then you think this is a little bit. So I don't know [if they are useful to me]"* (Nurse C) *"No, because none of those sentences have anything in it. It does not say how many patients were there, it does not say how many activities were completed, it's just many and most. [. . . ] How much, how many? It doesn't say much."* (Nurse C).

Doctors and nurses both recognize that linguistic summaries may create a better awareness of compliance among all staff. They also acknowledge the insights for management that may be more accessible through the use of the summaries: *"I think from managerial perspective it would be useful to use linguistic summaries, because on a higher level of management (in our case from ICU you go to the [medical] center) then you talk to people like Frits XXX [i.e., executive board member] who is not a doctor, but who is a manager. The perception how to discuss this is different I think."* (Doctor D). *"Maybe, [this information would be useful] to trigger the use of it. To see that only 40% of nurses use the protocol, that they would maybe think, oh, we really need to start using the protocol more. So I think that would be helpful, to get updates every now and then, and for your manager to check if we are using it enough and if we are using it good as well. It wouldn't be bad."* (Nurse C).

## 6. Discussion

During the interviews, we noticed that while all interviewees find protocol compliance very important, the meaning of the term "protocol compliance" is not well defined, and every interviewee had a different interpretation of what compliance with the protocol means to them. One of the doctors mainly reasoned at the department level, and the other at the patient level (how many of the treatment actions completed for one patient are according to the protocol), while one of the nurses mainly wanted to know whether she personally complied with the protocol (i.e., whether the actions she took were according to the protocol), and the other nurse as to whether the activities performed with the patients were compliant with the protocol. While the term protocol compliance in theoretical definitions refers to whether the process described in the protocol is followed for a patient, we noticed different interpretations on different aggregation levels. *"[. . . ] We approach the issue of the evaluation of the protocol or guideline always from a patient perspective assuming that the protocol is properly constructed."* (Doctor W) *"[. . . ] Obviously managers are interested in glucose*

*determinations we do, because it cost money for us. And these people do not understand medical terminology. So linguistic summaries for these groups are more easy to understand as well [. . . ] If you approach it from value-based healthcare perspective, so if you want to contribute to the increased or same level of care with decreased level of cost, then for these managers linguistic summaries are easier to understand. Depends on which level of management you consider."* (Doctor W).

From this research, we can conclude that the topic of compliance is interesting yet remains an open topic to be explored in more depth. We noticed that there are three independent dimensions to analyze the protocols, as shown in Figure 5, namely:

- Regulation perspective (action vs. protocol perspective).
- Object perspective (single patient vs. cohort of patients).
- Resource perspective (individual nurse vs. department).

Those dimensions can have different granularity levels, and they can be combined together, e.g., analyzing all young nurses' activities if a certain protocol is used for a certain group of patients.

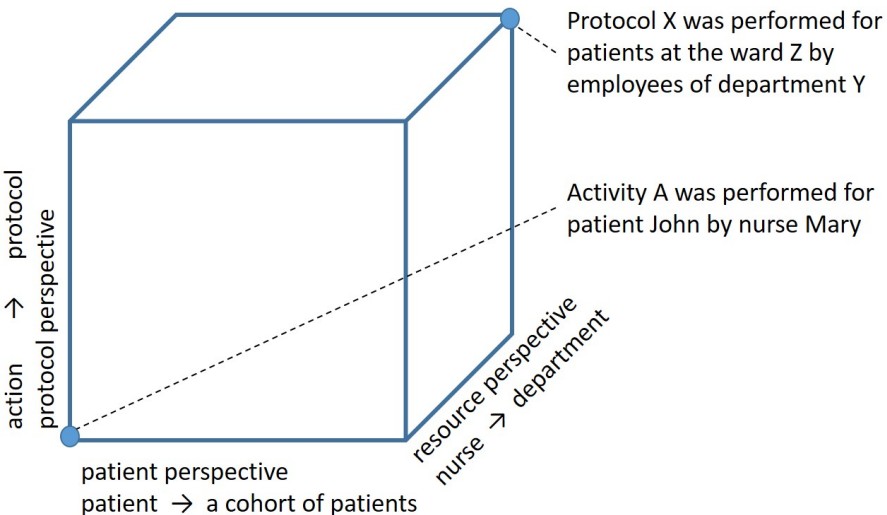

**Figure 5.** Three dimensions for a protocol compliance analysis.

One more interesting observation is the role of the experience of adhering to the protocol. It seems that more experienced nurses (are allowed to) deviate more from a protocol than novice nurses. This could result in flexible (or dynamic) membership functions for compliance for individual healthcare providers.

An issue not considered here is the interaction of the protocols. For instance, in the case of the glucose management protocols, medications administered to the patient can influence glucose levels. Therefore, administering those medications can influence the optimal insulin dosage for the patients. Currently, such cases are not included in the protocols. Including all cases where there may be a reason for deviation is unfeasible; therefore, allowing and capturing deviations from the protocol is a simpler option.

## 7. Concluding Remarks

In this paper, the problem of assessing compliance with the clinical protocols was addressed. Our work in this paper contributes to the efforts that fall under the ex-post (offline) compliance assessment approaches, while the proposed solutions can also be adapted for design-time and run-time compliance assessments. We consider the concept of compliance as a matter of degree; therefore, we use fuzzy sets to map it, similarly to [13,19]. We used two fuzzy aggregation techniques to calculate the compliance measures: OWA operator and Sugeno integral. The proposed measures take into consideration three factors: a single activity can be compliant with a degree, a patient, and the importance of the activities. Proposed measures were applied to two clinical protocols used in practice:

glucose management protocol and weaning protocol. Measures for compliance, such as those proposed, could further aid clinicians in evaluating the aspect of protocol compliance when evaluating the effectiveness of implemented clinical protocols. Evaluations show that compliance is not a static concept, and the level of flexibility could depend on, e.g., experience. Moreover, different perspectives, such as a single resource perspective, could be beneficial to improving the overall quality of care. Furthermore, compliance measures, although informative, are not yet sufficiently actionable. Hence, future solutions should include improvement guidance. In future work, we plan to investigate different interpretations of unknown information on the protocol compliance. The second aspect, not addressed in this paper, is a log-protocol alignment. There are many methods available for this purpose, e.g., [50,79]. Future work will analyze the influence of this aspect.

**Author Contributions:** Conceptualization, all authors; methodology, all authors; data analysis, A.W.; evaluation A.W. and I.V.; writing—original draft preparation, A.W. and I.V.; writing—review and editing, all authors. All authors have read and agreed to the published version of the manuscript.

**Funding:** This research received no external funding.

**Institutional Review Board Statement:** Not applicable.

**Informed Consent Statement:** The Research Ethics Review Committee (ERC) considers this research 'Evaluation of regular care', and therefore, patient consent was waived.

**Data Availability Statement:** Not applicable.

**Acknowledgments:** We thank J. Townsend for assistance with data collection and R. Lips, I. Kuiper, A. Reijers for assistance with data processing. We thank C. Eleveld, U. Kaynak and R. Kusters for comments and discussions around the compliance topic.

**Conflicts of Interest:** The authors declare no conflict of interest.

## Appendix A. Questions Asked during the Evaluation Interviews

1. How would you define compliance?
2. Is compliance to clinical protocols and guidelines important/used in ICU?
3. To what extend small deviations from the protocol/guideline are acceptable?
4. Is it more important to assess compliance with respect of individual activities or per patient? (e.g., better to have 60% compliant activities or 60% compliant patients?)
5. To what extend statements like:

   - "many patients have almost all of their activities compliant"
   - "most patients have most of their activities compliant"
   - "almost all patients have some of their activities compliant"

   are comprehensible/useful for you? Can they help you in your work?
6. Are some activities more important than the others in the clinical protocols?
7. To what extend statements like:

   - "many patients have almost all of the important activities compliant"
   - "most patients have most of the important activities compliant"
   - "some patients have almost all of the important activities compliant"

   are comprehensible/useful for you? Can they help you in your work?
8. To what extend a number like 70% denoting "at least 70% of patients have 70% of the activities compliant" is comprehensible/useful for you? Can this help you in your work? item Is such information useful from managerial perspective?

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
