# Peer review of "Towards a Flexible Assessment of Compliance with Clinical Protocols Using Fuzzy Aggregation Techniques"

_algorithms, doi:10.3390/a16020109_

Round 1
Reviewer 1 Report
Application of two types of aggregation operators in medicine, i.e., in real life problems , is a really interesting and up to date topic.
Please, rewrite mathematical part in a more formal way.
You are using notion "fitness measure". Please explain what type of measure (in mathematical sense) that is.
Difference between (4) and (10) (or (2) and (8)) is not well explained. What is the difference between the input values?
\lambda-fuzzy measure needs some more attention. For a fixed \lambda you construct the corresponding \lambda measure. If \lambda=0, you have additivity. Sugeno that you are observing is in a discrete setting, that is why set X is finite. \lambda is not determined by (14), (14) follows from assumption that g is \lambra-fuzzy measure from some given \lambda....
This whole part needs some more attention.
Author Response
Reviewer 1 had three main comments:
1) fitness measure – it is a measure as it satisfies the properties. Fitness can be interpreted as a distance between a log trace and a closest execution path.
2) Difference between OWA operator and OWA operator with importance qualification is how the weights are defined. Redundant parts were removed
3) \lambda fuzzy measure – if \lambda is equal to 0 we have a probability measure. Otherwise, if the densities are known, we can derive \lambda from (14) and can calculate the value of the fuzzy integrals. This way of defining fuzzy measure was often used by Keller and his collaborators. The fuzzy measure definition was improved.
Reviewer 2 Report
The paper proposes new metrics for compliance assessment in healthcare using fuzzy aggregation techniques.
Overall, the paper is interesting and wants to solve a real problem. However, to improve the quality of the work, I suggest the following improvements.
The authors choose to formalize the linguistic variable with a membership function of trapezoidal type.
However, it is necessary to explain the motivation behind this choice. Please, the authors explain.
Also, the fuzzy rules have not been shown. It would be interesting to understand how they made it and how many there are. This aspect is fundamental to understanding the comprehension of the model. Please, the authors show the knowledge base.
The authors should better explain how the fuzzy model was composed. Are the experts who formalized it the same ones who evaluated it?
Finally, I suggest considering other fuzzy methodologies to compose the knowledge base or to apply fine-tuning of the model.
For example, these works consider neuro-fuzzy models:
· G. Casalino, G. Castellano, U. Kaymak and G. Zaza, "Balancing Accuracy and Interpretability through Neuro-Fuzzy Models for Cardiovascular Risk Assessment," 2021 IEEE Symposium Series on Computational Intelligence (SSCI), 2021, pp. 1-8, doi: https://doi.org/10.1109/SSCI50451.2021.9660104.
· Karaboga, D., Kaya, E. Adaptive network based fuzzy inference system (ANFIS) training approaches: a comprehensive survey. Artif Intell Rev 52, 2263–2293 (2019). https://doi.org/10.1007/s10462-017-9610-2.
These models, through a data-driven approach, build a fuzzy knowledge base. I suggest the authors consider them as future work.
Author Response
Reviewer 2 has two main comments:
1) Motivation why trapezoidal membership function was used. Motivation is given in the paper in lines 221-226.
2) The reviewer asks about fuzzy rules. In the paper there are no fuzzy rules, in the sense of inference systems. What we do in this paper is: we calculate the discrepancies between the rules of the protocol and the execution, next we assess the degree to which the discrepancy is noncompliant and as the last step we aggregate those degrees of noncompliance using OWA operators and Sugeno integral.
Reviewer 3 Report
A novel fuzzy aggregation metrics applied for clinical protocol compliance assessment is proposed.
This paper is well written and well structured and the proposed research is innovative and interesting.
For greater readability of the text, I suggest to the authors to add brief descriptions of the benefits of the proposed model and how it solves the critical points of the models found in the literature presented in the section.
In section 3 it is useful to schematize the proposed methods in algorithmic form in pseudocode by numerically recalling the single equations. This, in fact, offers a clearer and more structured picture of the methods of applying the aggregation metrics to the reader.
In the concluding section it is necessary to highlight in more detail the critical and unexplored points of the compliance assessments, etrics proposed, linking this description to the future prospects of the research.
Author Response
The reviewer 3 has three main comments:
1) The advantages of the proposed methods were added to the appropriate sections.
2) The algorithms pseudocodes were added
3) Conclusion section was extended to highlight critical and unexplored points of the compliance assessments